# Realization of acoustic spin transport in metasurface waveguides

Yang Long [1,3], Danmei Zhang[1,3], Chenwen Yang[1], Jianmin Ge[2], Hong Chen[1] & Jie Ren [1✉]

Spin angular momentum enables fundamental insights for topological matters, and practical implications for information devices. Exploiting the spin of carriers and waves is critical to achieving more controllable degrees of freedom and robust transport processes. Yet, due to the curl-free nature of longitudinal waves distinct from transverse electromagnetic waves, spin angular momenta of acoustic waves in solids and fluids have never been unveiled only until recently. Here, we demonstrate a metasurface waveguide for sound carrying non-zero acoustic spin with tight spin-momentum coupling, which can assist the suppression of backscattering when scatters fail to flip the acoustic spin. This is achieved by imposing a soft boundary of the $\pi$ reflection phase, realized by comb-like metasurfaces. With the special-boundary-defined spin texture, the acoustic spin transports are experimentally manifested, such as the suppression of acoustic corner-scattering, the spin-selected acoustic router with spin-Hall-like effect, and the phase modulator with rotated acoustic spin.

---

[1] Center for Phononics and Thermal Energy Science, China-EU Joint Center for Nanophononics, Shanghai Key Laboratory of Special Artificial Microstructure Materials and Technology, School of Physics Science and Engineering, Tongji University, Shanghai 200092, China. [2] Institute of Acoustics, Shanghai Key Laboratory of Special Artificial Microstructure Materials and Technology, School of Physics Sciences and Engineering, Tongji University, Shanghai 200092, China. [3]These authors contributed equally: Yang Long, Danmei Zhang. ✉email: xonics@tongji.edu.cn

The spin angular momenta (SAM) of classical waves, ranging from elastic to optical waves, are essentially associated with their local chiral polarized profiles. Many well-known waveforms, such as circularly polarized plane wave, two-wave interference, spatial confined Gaussian beam, and surface evanescent wave, have been shown to possess nonzero SAM[1–8]. The SAM exhibits a family of spin-related effects with remarkable properties for waves, i.e., spin-Hall effect (SHE)[2,9], quantum spin Hall effect (QSHE)[10,11] and spin-locked scattering in bianisotropic media[12]. Meanwhile, these findings give rise to some emergent classes of research fields, like the chiral quantum optics[13]. Many applications have then been proposed to control classical waves with SAM, such as spin-selective plasmonics excitations[14], chiral dependencies of quantum optical devices[15], hyperbolic metamaterial wave routers[5,16], and so on. The advantages of robustness and flexible control of the spin-based wave devices would have impact on many areas of wave theories, experiments, and devices[17–25].

In the past, vast previous research focused on the orbital angular momentum of acoustics[26] without thinking about the existence possibility of SAM due to the curl-free nature of longitudinal waves. Because conventionally it is enough to represent the acoustic wave by scalar pressure fields only, and classical field theory concludes that scalar fields possess zero SAM[27]. Therefore, people widely believed that the curl-free longitudinal wave cannot shape into circularly polarized from as transverse optical waves so that the acoustic waves cannot possess SAM. Just recently, after fully treating the acoustics as vectorial velocity fields, the fact that the acoustics can process the SAM intrinsically has been unveiled theoretically and observed experimentally[5–8]. The acoustic SAM can thus be naturally associated with the circularly (elliptically, in general) polarized profile of acoustic velocity fields (nonzero $\mathbf{v}^* \times \mathbf{v}$), which does not conflict with the zero vorticity $\nabla \times \mathbf{v} = 0$ of the curl-free acoustic waves. After these improvements in understanding the longitudinal wave SAM, exploiting the acoustic SAM in practice becomes of significance and attracts considerable attention, including spin–orbit coupling in acoustic Bessel beams[28], acoustic spin-induced torque[29]. In this work, we propose metasurface waveguides to demonstrate several acoustic SAM applications on spin-related robust transport. The key of our work is based on the nonzero acoustic SAM resulting from the specific waveguide modes after introducing metasurface boundary conditions. As shown in Fig. 1a) we find that the SAM of modes is strongly coupled to the propagation direction (momentum), resembling the relation of spin-momentum locking in QSHE states[2,25]. Owing to these properties, the acoustic waveguide mode exhibits backscattering-suppressed transport if scatters do not flip the SAM, and becomes spin-selective when facing the multiple channels due to different SAM-momentum dependences. We perform several experiments to exemplify these spin-related phenomena. Finally, we obtain the rotated acoustic spin as a phase modulator induced by gradually rotating the meta-waveguide boundary. Our work would give more fundamental understandings about wave spin physics and provide new insights about controlling waves via SAM degree of freedom.

## Results

**Nonzero acoustic spin in waveguide with symmetry breaking**. According to the physical meanings possessed in angular momenta of acoustic waves, the SAM density can be described as (refs. [5–8]):

$$\mathbf{s} = \frac{\rho}{2\omega} \text{Im}[\mathbf{v}^* \times \mathbf{v}], \tag{1}$$

where $\rho$ is the mass density, $\omega$ is the frequency, $c$ is acoustic velocity, and $\mathbf{v}$ is acoustic velocity field. According to the

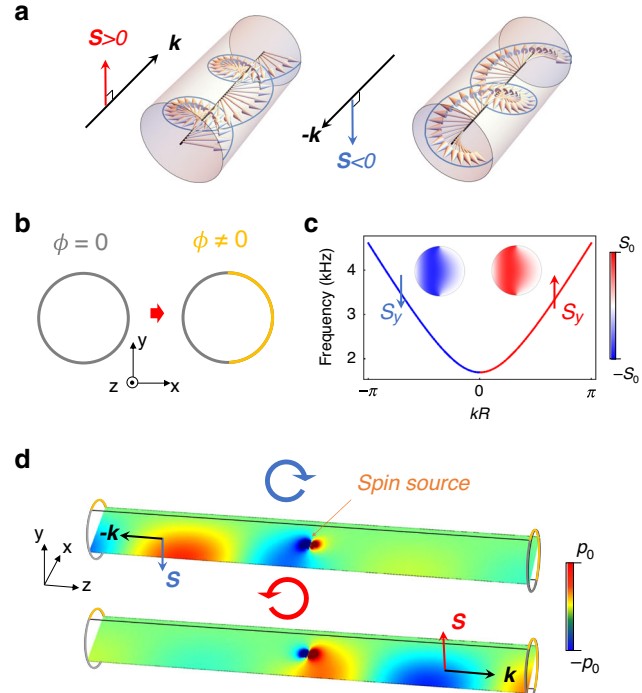

**Fig. 1 Spin angular momentum in waveguide with symmetry-breaking boundary conditions. a** The schematic description of spin-dependent propagations in waveguide. The circular evolutions of velocity fields (Gray arrows) in propagating acoustic waves reflect the nonzero transverse spin angular momentum according to the acoustic SAM definition in Eq. (1). **b** Cross-sectional view of acoustic waveguide with nonsymmetric boundary conditions with different reflection phase $\phi \neq 0$. **c** The spin-resolved dispersion along z-direction. We use the red/blue color to represent the sign of acoustic SAM of eigenmodes $S_y = \int s_y d\mathbf{r}^2$, red for $S_y > 0$ and blue for $S_y < 0$. The radius of waveguide is $R = 4$ cm. The reflection phase is set as $\phi = \pi$ (metasurface boundary). **d** The selective excitation of these waveguide modes can be realized by exploiting the spin sources (circularly polarized acoustic dipoles) placed in the center of waveguide.

definition in Eq. (1), the acoustic SAM density will be nonzero for the local circularly (or in general, elliptically) polarized profile of acoustic velocity fields. To satisfy the circularly polarized velocity field conditions, we introduce nonsymmetric boundary settings to break the x-mirror symmetry of waveguide, shown in Fig. 1b, which will led to significant phase delay for $v_x$ compared with $v_z$. By general symmetry argument, setting $\mathcal{M}_y(y \to -y)$ as the y-mirror operation, we have $\mathcal{M}_y s_x(y) \mathcal{M}_y^{-1} = -s_x(-y)$, $\mathcal{M}_y s_y(y) \mathcal{M}_y^{-1} = s_y(-y)$ and $\mathcal{M}_y s_z(y) \mathcal{M}_y^{-1} = -s_z(-y)$. So for the nondegenerate waveguide mode, when the cross-section has the y-mirror symmetry, the cross-section SAM $S_x = \int s_x dx dy$ and $S_z = \int s_z dx dy$ will definitely vanish due to the cancellation of odd-symmetric SAM densities on the waveguide section. Similarly, the x-mirror symmetry of the system will lead to the vanishing $S_y = 0$ and $S_z = 0$ due to the odd-function $s_{y,z}(x) = -s_{y,z}(-x)$. Therefore, to induce the nonzero SAM, e.g., $S_y \neq 0$ for nondegenerate modes, one could break the $\mathcal{M}_x$ symmetry.

Considering the combination of the sound hard boundary ($\phi = 0$) and the metasurface-induced soft boundary ($\phi = \pi$) in Fig. 1b, we can analytically obtain that the waveguide mode indeed processes the nonzero total SAM as (see "Method"):

$$\mathbf{S} = \frac{\pi |p_0|^2}{\rho \omega^3} kR J_0^2(\kappa R) \mathbf{e}_y, \tag{2}$$

where $J_0(\kappa r)$ is the zero-order Bessel function of the first kind, $\mathbf{e}_y$

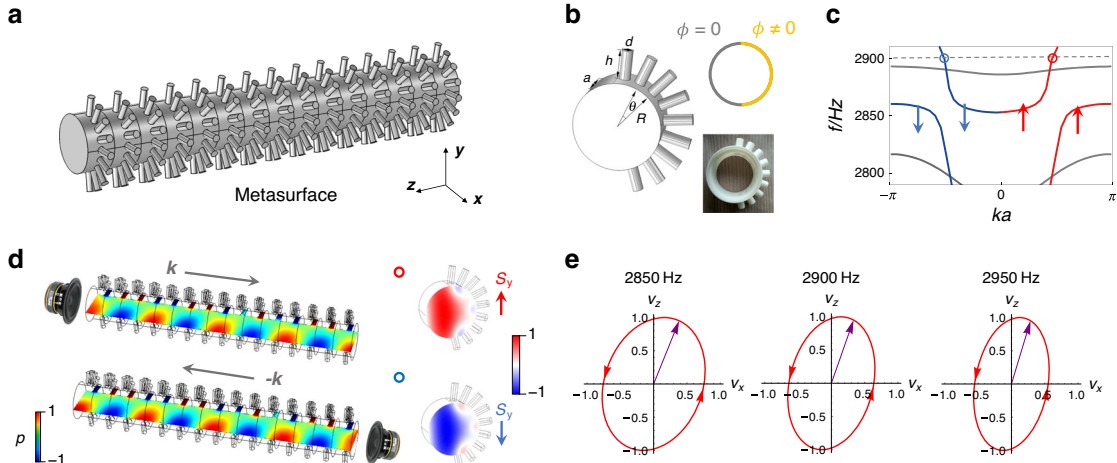

**Fig. 2 Spin angular momentum in acoustic waveguide with metasurfaces. a** The side bar structures are exploited to achieve the acoustic metasurface with arbitrary reflection phase $\phi \neq 0$. **b** The unit cell of metasurface waveguide. Here, we choose the geometrical parameters as: $a = R = 4$ cm, $d = 1$ cm, $h = 2.5$ cm, $\theta = 20°$, to achieve $\phi = \pi$. **c** The $z$-direction dispersion relationship is calculated and the sign of tranverse SAM $S_y$ in eigenmodes have been shown by the red and blue colors. **d** The excited waveguide modes with different propagation directions will carry opposite SAM texture at 2.9 kHz. **e** The elliptically polarized velocity field simulated at the center point of the metasurface waveguide. The plotted velocity field is normalized by the amplitude of $v_z$. The purple arrow and the arrowed red circle denotes the polarization of the velocity field and its time evolution (More cases can be found in Supplementary Note 2).

is the unit vector in $y$-direction, $k$ is the longitudinal momentum component with $\mathbf{k} = k\mathbf{e}_z$ the wave vector along $z$-direction, $\kappa = \sqrt{\omega^2/c^2 - k^2}$ is the transverse wave vector, $c$ is the sound speed in the air, $R$ is the radius of the waveguide, and $p_0$ is the pressure field amplitude. Based on these nonsymmetric boundary conditions, the meta-waveguide eigenmodes will naturally carry nonzero SAM $S_y \neq 0$ in the $y$-direction, depicted in Fig. 1c. Specially, this SAM is tightly locked to the momentum direction that reversing the momentum $k$ will flip the spin, which resembles the spin-momentum locking in QSHE[2,11]. By using the spin sources (circularly polarized acoustic dipoles)[5,6] corresponding to different SAM, we can excite the waveguide modes selectively, as shown in Fig. 1d.

It should be mentioned that this spin-momentum locking feature of meta-waveguide modes is different from the cases in the surface evanescent modes[1,30] due to nontrivial bulk topology and bulk-edge correspondence[25,31], where the field strength is strongly localized at (meta-)surfaces. For our waveguide modes, the SAM-distinguished dispersion is attributed to the structure-dependent wave interference and the resulting chiral velocity field wrapped by the phase-delayed metasurface. Specifically, the acoustic wave is spatially confined well in the area near the normal surface but severely attenuated in the area near the metasurface (see Fig. 1c, d), the latter of which is caused by the destructive interference due to the phase-delayed reflective metasurfaces. This transverse attenuation induces effective chiral fields perpendicular to the waveguide propagation direction, similar to the case in Gaussian longitudinal wave beams[5].

**Acoustic metasurface structures**. We exploit the side bar structures to realize arbitrary phase-delayed reflective acoustic wave metasurfaces, as shown in Fig. 2a, b. The side bar arrays on the waveguide boundary can be regarded as acoustic meta-atoms[32]. These meta-atoms can reflect the acoustic wave with arbitrary phase delay when they become resonant[33]. As such, we can find that the waveguide mode can carry nontrivial acoustic SAM $S_y \neq 0$, and the SAM is strongly associated with the linear momentum responsible to the propagation shown in Fig. 2c. This

momentum-dependent SAM will led to the opposite SAM texture of modes for different momentum excitations in Fig. 2d. Specially, the SAM densities is associated with the near-circularly (elliptically) polarized velocity fields. As demonstrated in Fig. 2e, metasurface waveguide modes with $k > 0$ at different frequencies will result in the top-viewed anticlockwise elliptically polarized velocity fields corresponding to $s_y > 0$, which make opposite propagating modes of opposite polarizations become approximately orthogonal to each other. This momentum-locked SAM profile of bulk modes within the waveguide will apparently reduce couplings between forward propagating and reflected scattering modes. To experimentally verify the SAM densities inside the metasurface waveguide, we perform experimental measurements (see "Method") and compare them with analytic theory predictions Eq. (2) and numerical simulations, as shown in Fig. 3. One can see that the experimental measurements are in good agreements with theoretical and numerical results. Indeed, the $y$-mirror symmetry makes the odd-symmetric $s_x$ with $s_x(y) = \mathcal{M}_y s_x(y) \mathcal{M}_y^{-1} = -s_x(-y)$, resulting in the vanishing $S_x = \int s_x dx dy = 0$ due to the cancellation in the integral. However, the $x$-mirror symmetry breaking leads to the non-perfect cancellation $s_y(x) \neq -s_y(-x)$, facilitating the nonzero $S_y \neq 0$.

**Experimental observations of acoustic spin transport**. With this special-boundary-defined spin texture and the inherent spin-momentum locking shown in Eq. (2) and Fig. 3, the transport in the metasurface waveguide would be robust compared with conventional spinless waveguide modes. As demonstrated in Fig. 4a, we bend the metasurface waveguide with the bending angle $\theta$. In the Fig. 4b, the simulated transmission $T$ of the bended metasurface waveguide will become better than the conventional circular duct waveguide, especially for $\theta \in [0, \pi/2]$, which will be nearly 100% transmission. This is assisted by the tight acoustic spin-momentum locking of the acoustic spinful mode in the meta-waveguide (see Eq. (2) and Fig. 2), which for opposite transport directions (momentum) has opposite SAMs. In particular, the nonsymmetric transmission $T(\theta) \neq T(-\theta)$ can be found for the metasurface waveguides with opposite bending angles, which indicates that the spinful waveguide mode will pass more

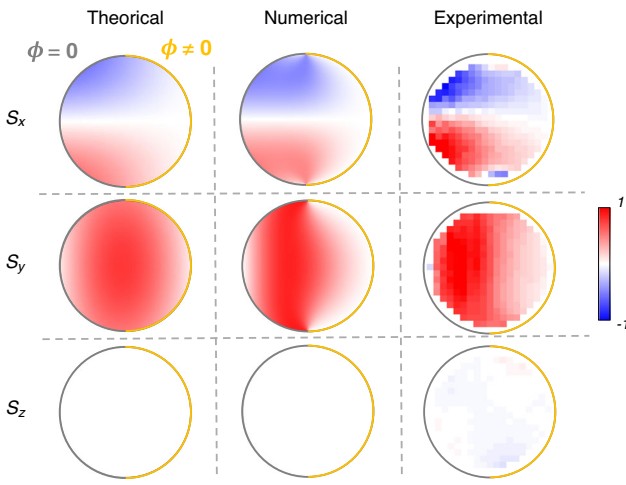

**Fig. 3 Comparison among theoretical, numerical and experimental results for SAM density on the waveguide cross-section.** Theoretical results, numerical simulations and experimental measurements coincide with each other very well. The theoretical results of SAMs are from the analytic expression Eq. (2). The experimentally SAM density profile $s_y$ is measured for $k > 0$ in the frequency 2.9 kHz (Red circle cases in Fig. 2(c)).

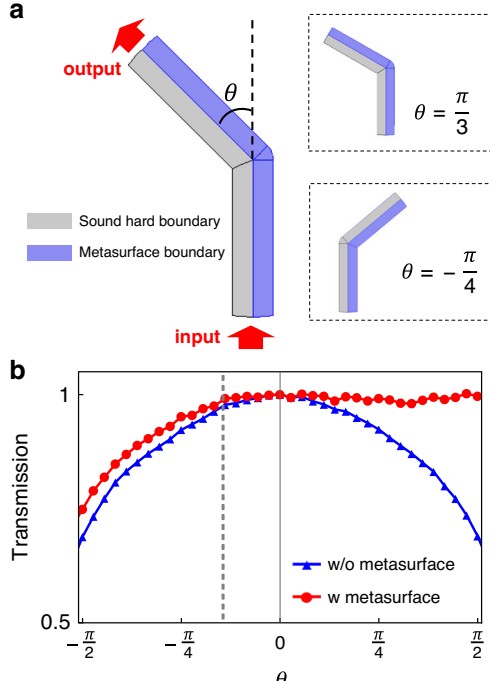

**Fig. 4 Bending effects on the acoustic spin transport. a** The bended metasurface waveguide with the bending angle $\theta$. Two bending cases are shown for clarifying bending structures with a corner defect. The sound hard boundaries and ideal metasurface boundaries with the reflection phase $\pi$, are denoted as the gray and blue color, respectively. **b** The transmissions of the metasurface waveguide (red dotted line) and the conventional circular duct waveguide (blue dotted line) as the function of the bending angle $\theta$. The side bar structures of metasurfaces will have volume overlaps when $\theta < -0.14\pi$ (gray dashed line). The demonstration frequency is 2.9 kHz.

easily through corners decorated by metasurfaces. This might be attributed to the asymmetric distribution of the waveguide field, i.e., the pressure fields mainly localize on one half of the waveguide cross-section without metasurface decorations. In the following, we are going to experimentally explore the acoustic spin transport in several typical examples, including robust transport, spin-selective routing and rotating spin under effective magnetic field.

*Robust transport with corner-scattering suppression due to tight spin-momentum coupling*: We first simulate the acoustic field of a U-shape normal acoustic waveguide without metasurface when incident wave imposed at one port. Results show that the strong backscattering will happen at bending corners, as depicted in Fig. 5a. As a contrast, we simulate the acoustic field when incident acoustic wave into the similar U-shape waveguide but with metasurface. From the simulation results shown in Fig. 5a, we can see the acoustic waves successfully pass through the U-shape meta-waveguide with low scattering loss at corners.

To verify these spin-related backscattering-suppressed effects, we perform experiments to measure the transport of U-shape acoustic waveguide with/without metasurfaces as shown in Fig. 5b. From the experimental results in Fig. 5c, we can see that the U-shape has little effects on the transmission of the metasurface waveguide at frequency region 2.85–2.95 kHz, which is exactly around the resonant frequency of the side bar structures. But in contrast, the waveguide without metasurface does not hold a clear acoustic SAM texture and it will suffer strong scatterings leading to severely attenuated transmission. Compared with conventional waveguides, this robustness transport of acoustic spinful mode in metasurface waveguides is assisted by the nonzero SAM and its tight spin-momentum locking relation, resulting in that the back-propagation requires spin flips to opposite one. Acoustic SAM textures represent polarization profiles of velocity fields. To ensure the spin (polarization) matching between opposite propagating acoustic modes of opposite SAMs, backscattering requires strong scatters that reverse spin. Indeed, scattering behaviors highly depend on the type of scatters and the interaction with wave modes. For example, certain types of impurities and defects will break topological transport protections in topological insulators without preserving time-reversal symmetry so that backscattering occurs

with spin flip[34,35]. In fact, many scatters in metasurfaces will possess spin-related scattered/reflected properties[14,36,37]. From experimental results, one can see that here the metasurface waveguide modes with nonzero SAM is insensitive to the present corner defects.

*Spin-selective wave router based on opposite acoustic-spin-dependences of different directions*: As discussed before, the momenta of metasurface waveguide modes are tightly coupled to the SAM of modes, which means that the wave will propagate along the direction selectively based on the SAM match, when facing multiple transport channels with opposite SAMs. To observe the selective phenomena, we design a T-shape structure and confirm the spin-selective router by simulation results as shown in the Fig. 5d. From Fig. 1, we know that the acoustic mode to the left export is the same up-spin state ($S_y > 0$) as the incident acoustic wave, while the right export has opposite down-spin state, so that the acoustic wave will choose the left direction with the same up-spin. If we rotate the incident waveguide around incident axis for 180°, the incident acoustic wave will possess a flipped SAM. The right export will be the same down-spin state ($S_y < 0$) as the incident acoustic wave, while the left export has opposite up-spin state. As such, the acoustic wave now will choose the right direction with the same down-spin as expected. This is reminiscent of the spin-Hall-like effect for electrons and optics[36,37].

Experimentally, we input the acoustic wave from the port and measure the transmission at sides A/B shown in Fig. 5e. The experiment results in Fig. 5f show that the acoustic wave chooses one side (A) to propagate and leave no output for the other one

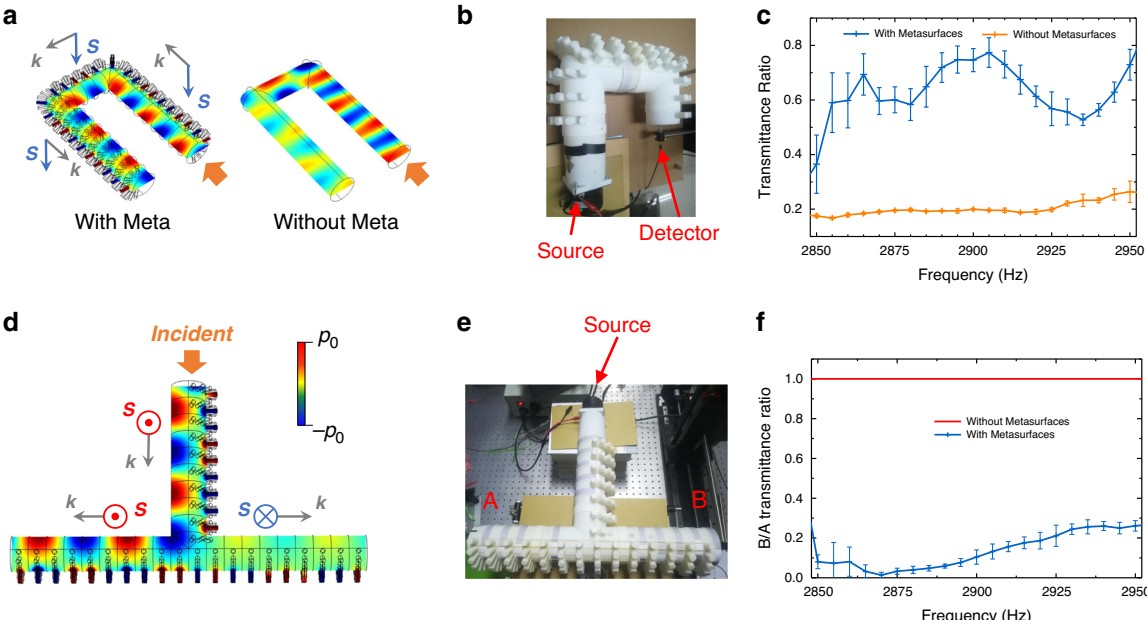

**Fig. 5 Acoustic SAM-induced robustness on wave transport and SAM-related wave routing. a** The robustness can be reflected by the transmittance ratios of waveguides with/without metasurfaces to the unperturbed straight ones. The pressure fields are plotted in the normalized color bar. **b** The experimental settings. The source excites the waveguide modes and the pressure fields are measured by the detectors. The transmittance ratios are obtained by calculating the ratio between the U-shape waveguide and straight waveguide with the same length. **c** The experimental results show that the nonzero SAM of acoustic modes and the associated tight spin-momentum coupling will enhance the backscattering suppression. **d** The acoustic wave will selectively propagate along one side due to the SAM match when facing multiple channels. Reversing the incident acoustic wave spin will change outgoing waves to select the other channel with opposite spin. This acoustic spin-selective directionality is reminiscent of the spin-Hall-like effects for electrons and optics. **e** The experimental results to demonstrate the selective transport based on the matched SAM. The pressure field strengths at the port A and B are recorded. And the ratio between the port A and B will reflect this SAM-based selectivity. **f** The experimental results (blue line) show the strong selective phenomena. The red line means that the wave transport is divided equally in the normal T-waveguide without metasurfaces, namely $A = B$.

(B). At the crossing point, the SAM of modes will make the acoustic wave propagate along the direction with the matched SAM. Moreover, it shows that this spin-selective transport will also be insensitive to the backscattering at T corner, due to the asymmetric spatial distribution of acoustic spinful waveguide modes and the assistance of spin-momentum locking.

*Gradually rotated metasurface waveguide for rotating acoustic spin and phase modulator*: As is known from Fig. 1, the acoustic wave polarization confined within the meta-waveguide forms complicated SAM texture, which highly depends on the boundary geometry settings of metasurfaces. This indicates that following a gradually rotating meta-waveguide boundary, the evolving polarization of acoustic wave will result in rotating SAM textures, as shown in Fig. 6a. By rotating metasurface boundary with an angular velocity $\Omega$ along $z$-axis, we find that the variation of acoustic SAM texture in $xy$ plane will follow the rotating evolution of velocity polarized profile $(\frac{\partial}{\partial z}\mathbf{e}_i = \Omega \times \mathbf{e}_i)$ as: $\frac{\partial}{\partial z}\mathbf{S} = \Omega \times \mathbf{S}$. This is a reminiscent of "acoustic spin Bloch equation" that resembles the equation of motion of electron spin procession under an effective "magnetic field" $\Omega$[38]. The $z$-axis plays the role of a pseudo-time so that the wave propagation along this pseudo-time will experience effective Larmor precession for acoustic spin with the Larmor frequency $\Omega$.

As the consequence of this effective Larmor precession of acoustic spin, the total acoustic wave phase accumulated during the propagation in rotating metasurface waveguide will have contributions from two individual phases $\Theta = \Theta_0 + \gamma$: the unperturbed phase $\Theta_0 = kL$ related with waveguide length $L$ and wave vector $k$; and the additional phase $\gamma = \frac{kR_0^2}{2L}\theta^2$ induced by adiabatic variation of meta-boundary conditions, where $\theta = \Omega L$ is

the final rotation angle (See Supplementary Note 3). To experimentally observe the phase $\gamma$ induced by rotating SAM texture, we rotate the boundary settings of metasurface waveguide slowly along $z$ direction and measure the corresponding transmitted phase, as shown in Fig. 6b at frequency $f = 2.85$ and 2.9 kHz, respectively. The measured phases $\gamma$ with different rotating rate $\Omega$ after the pseudo-time $L$ are in good agreement with theoretical results. Results indicate that the gradually rotating metasurface waveguide is also a phase modulator that has implications in acoustic interferometers.

## Discussion

To summarize, by introducing the special metasurface boundary to induce polarization-dependent phase delay, we have realized the nonzero acoustic SAM and associated tight spin-momentum locking. Owing to these properties, the acoustic waveguide mode exhibits backscattering-suppressed transport if scatters do not flip the SAM. Besides, the wave transport becomes spin-selective by choosing the right channel with matched SAMs when facing multiple channels with different SAMs, i.e., a spin-Hall-like feature. Finally, we have demonstrated a rotating metasurface waveguide and measured the additional phase modulation, resembling the spin precession under an effective magnetic field. Our work demonstrates spin-related acoustic transport mechanisms within the bulk, rather than at the interface[5,6]. This pure spinful bulk mode is determined by the special boundary with nontrivial reflection phase. Our results would supply new insights about the spin mechanism of vibrational waves and pave the way about acoustic SAM-related wave control, especially in the on-chip phononic devices[39] and even coupling to electron spin and quantum magnetism.

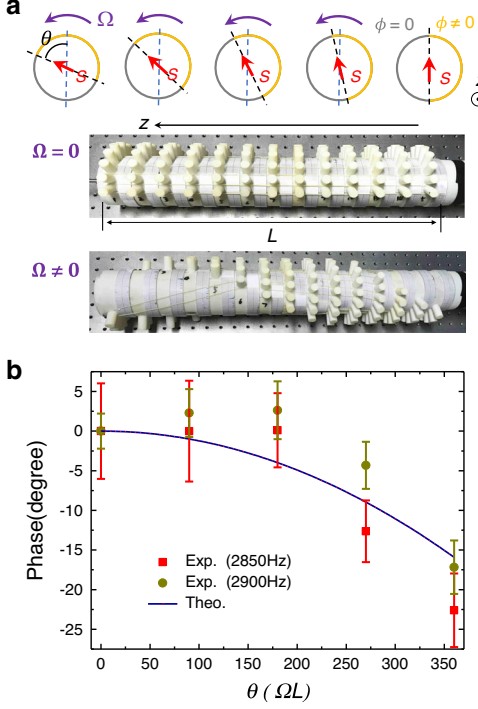

**a**

**b**

**Fig. 6 Rotating metasurface waveguide for rotating acoustic spin as a phase modulator. a** Rotating the boundaries of metasurface waveguide with angular velocity $\Omega$ will lead to rotating spin texture that resembles the motion of electron spin precession under an effective "magnetic field" $\Omega$. The final rotation angle will be $\theta = \Omega L$, where $L$ is the length of waveguide. **b** The experimentally measured acoustic additional phase $\gamma$ at 2.85 and 2.9 kHz with the standard deviation as the error bar. The theoretical prediction can be represented by $\gamma = \frac{kR_0^2}{2L}\theta^2$, where $k = 24$. 5°/cm is the measured wave vector for 2.9 kHz, $L = 60$ cm and $R_0 = -1.4$ cm (See Supplementary Note 3).

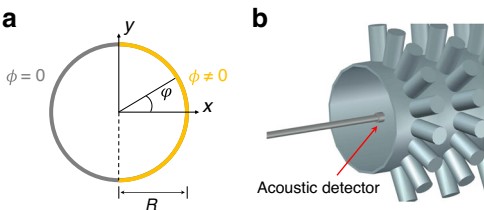

**a**                    **b**

**Fig. 7 The coordinate and the experimental scheme. a** The polar coordination in the metasurface waveguide. The gray and yellow color denote the conventional sound hard boundary and metasurface boundary respectively. Here, we consider the reflection phase of the metasurface is $\phi = \pi$ (corresponding to the sound soft boundary). **b** the experimental scheme to detect the SAM density. The acoustic detector is inserted into the waveguide, moved by the autostage and recording the acoustic field.

## Methods

**Spin angular momentum in the metasurface waveguide**. In the cylindrical coordination system $(r, \varphi, z)$ as shown in Fig. 7a, the solution of the acoustic field in the metasurface waveguide mode will be the linear composition of the Bessel function $J_l(r)$ as: $p = p_0\sum_l c_l J_l(\kappa r)e^{il\varphi}e^{i(kz-\omega t)}$, where $c_l$ is the modal amplitude, $\kappa = \sqrt{\frac{\omega^2}{c^2} - k^2}$, $\mathbf{k} = k\mathbf{e}_z$ is the longitudinal wave vector along the $z$ direction, $\omega$ is the frequency and $c$ is the sound speed in the air. Due to the nonsymmetric boundary conditions in Fig. 7a, we will have: $p_{r=R} = 0$ for $\varphi \in [-\pi/2, \pi/2]$ and $\partial_r p|_{r=R} = 0$ for $\varphi \in [-\pi, -\pi/2) \cup (\pi/2, \pi]$, where $R$ is the radius of the waveguide. For the lowest order state for the metasurface waveguide, we consider only $l = 0, \pm 1$ here. After some maths, the acoustic pressure field for the ground state in the metasurface

waveguide can be approximately represented as:

$$p = p_0(J_0(\kappa r) + \alpha J_1(\kappa r)\cos(\varphi))e^{i(kz-\omega t)}, \quad (3)$$

with the dispersion $k^2 + \kappa^2 = \frac{\omega^2}{c^2}$, $\alpha$ is the constant. With the boundary conditions, we will have that: $\kappa R = \xi$, $\alpha = -\frac{J_0(\xi)}{J_1(\xi)}$, and $\xi$ is the first root of the equation $J_1^2(\xi) - \frac{1}{2}J_0(\xi)(J_0(\xi) - J_2(\xi)) = 0$ and $\xi > 0$. Thus, the velocity field $\mathbf{v} = -\frac{i}{\rho\omega}\nabla p$ will be :

$$v_r = \frac{p_0}{\rho\omega}\frac{i\kappa}{2}(2J_1(\kappa r) + \alpha(J_2(\kappa r) - J_0(\kappa r))\cos(\varphi))e^{i(kz-\omega t)}, \quad (4)$$

$$v_\varphi = \frac{p_0}{\rho\omega}\frac{i\alpha}{r}J_1(\kappa r)\sin(\varphi)e^{i(kz-\omega t)}, \quad (5)$$

$$v_z = \frac{p_0}{\rho\omega}k(J_0(\kappa r) + \alpha J_1(\kappa r)\cos(\varphi))e^{i(kz-\omega t)}, \quad (6)$$

and the corresponding SAM density $\mathbf{s} = \frac{\rho}{2\omega}\text{Im}[\mathbf{v}^* \times \mathbf{v}]$ will be:

$$s_r = -\frac{|p_0|^2}{\rho\omega^3}\frac{k\alpha}{r}J_1(\kappa r)(J_0(\kappa r) + \alpha J_1(\kappa r)\cos(\varphi))\sin(\varphi), \quad (7)$$

$$s_\varphi = -\frac{|p_0|^2}{\rho\omega^3}\frac{k\kappa}{2}(J_0(\kappa r) + \alpha J_1(\kappa r)\cos(\varphi)) \times (-2J_1(\kappa r) + \alpha(J_0(\kappa r) - J_2(\kappa r))\cos(\varphi)), \quad (8)$$

$$s_z = 0. \quad (9)$$

The $s_x$ and $s_y$ will be: $s_x = s_r\cos(\varphi) - s_\varphi\sin(\varphi)$, $s_y = s_r\sin(\varphi) + s_\varphi\cos(\varphi)$. As such, the total SAMs as the integral of acoustic SAM densities on cross-section $\mathbf{S} = \int \mathbf{s}d^2\mathbf{r}$ are obtained as:

$$S_y = \int_0^R \int_0^{2\pi} s_y r dr d\varphi = \frac{\pi|p_0|^2}{\rho\omega^3}kRJ_0^2(\kappa R), \quad (10)$$

and $S_x = S_z = 0$.

We note that from Fig. 3 one can find some unimportant mismatchings near boundaries between exact simulation results and analytical theory results. These are due to our lowest order truncation of the Bessel basis as approximations. When incorporating more higher-orders of Bessel functions, the theoretical field will coincide with the exact simulations very well (see Supplementary Note 1). Nevertheless, the present simplest approximation with lowest order up to 1 is sufficient to express the SAM values in a neat analytical form, which clearly uncovers the physics and reflects the key information: nonzero SAM with tight spin-momentum locking.

**Sample fabrication and experimental measurements**. All the elements are fabricated using 3D printing with the photosensitive resin material (modulus 2765 MPa, density 1.3 g/cm$^3$), and can be treated as acoustically rigid. The elements were 3D-printed, one unit cell at a time, and were connected to form different configurations. The 1/4-in. microphone (GRAS Type 46BE) is placed at the measuring terminal of the sample to detect amplitude and phase of the sound. The experimental environment was an open space to avoid unnecessary reflection. The detected signals are acquired by the NI 9234 data acquisition module. A sine signal is driven by the waveform generator (Keysight 33500B) with frequency range from 2.7 to 3.2 kHz. The sine signal was fed into a loudspeaker as the sound source. The sound waves emitted by the loudspeaker are guided into the tube. The position of the microphone is controlled by the automatic stage.

The acoustic field inside the waveguide is measured by placing the acoustic detector into the waveguide (the scale of the microphone is much smaller than the radius of the cross-section of the waveguide), moving the acoustic detector with the automatic stage and recording the acoustic pressure field $p_{\text{exp}}$, shown in Fig. 7b. The experimental SAM densities will be obtained by $\mathbf{s}_{\text{exp}} = \frac{\rho}{2\omega}\text{Im}[\mathbf{v}_{\text{exp}}^* \times \mathbf{v}_{\text{exp}}]$ and $\mathbf{v}_{\text{exp}} = -\frac{i}{\rho\omega}\nabla p_{\text{exp}}$.

The wave vector $k$ inside the waveguide is measured by moving the acoustic detector along the $z$-axis gradually and recording the phase change during the relocation. The wave vector $k$ will be the ratio between the phase change and the position change along $z$-axis. The additional phase detections are performed by recording phase data 3 times in each 3 points (with distances 6.5 mm, 16.5 mm, 26.5 mm to the center of the waveguide respectively) firstly and then averaging these detected phase data. Five rotation settings are experimentally prepared: $\Omega L = 0°$ (straight duct), 90°, 180°, 270°, and 360°.

## Data availability

The data that support the findings of this study are available from the corresponding author upon reasonable request.

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

## Acknowledgements

This work is supported by the National Key R&D Program of China (Grant No. 2016YFA0301101), the National Natural Science Foundation of China (No. 61621001, 11935010, and 11775159), the Natural Science Foundation of Shanghai (Nos. 18ZR1442800 and 18JC1410900), and the Opening Project of Shanghai Key Laboratory of Special Artificial Microstructure Materials and Technology.

## Author contributions

These two authors Y.L. and D.Z. contributed equally to this work. Y.L., D.Z., and C.Y. carried out the numerical simulations. D.Z., C.Y., and J.G. performed the experimental measurements. Y.L. and J.R. derived the theory and developed the analysis. J.R. and H.C. conceived the project. All the authors contributed to discussion, interpreting the data and the writing.

## Competing Interests

The authors declare no competing interests.
