## [Peer Review File · Nature Communications]

Reviewers' comments:

Reviewer #1 (Remarks to the Author):

This is an interesting paper, which reports an experimental observation of the robust acoustic-spin-controlled unidirectional transport in acoustic waveguides with metasurfaces. The presence of nonzero spin angular momentum density in structured sound waves was revealed very recently, and this direction of research was started by the authors of the current manuscript. Their present experimental demonstrations are novel and impressive. The topics of acoustic spin, metasurfaces, and unidirectional backscattering-immune transport are very hot, so this research is timely. However, I found a number of significant problems in this manuscript and interpretation of the results. Therefore, I will recommend publication of these results in Nature Communications only after the following questions are carefully considered and addressed by the authors, including some mandatory major changes in the manuscript.

1. The authors use the word "topological" in the title and throughout the paper. However, there is no topological theory supporting the phenomena observed here. There are only some symmetry-based arguments which involve the spin-direction locking in the modes of the metasurface-modified waveguide. (Note that the spin-momentum locking in electromagnetic/acoustic evanescent modes [3,15,20,24,29] has no clear topological origin, it should not be confused with truly topological phenomena involving bulk-boundary correspondence, topological numbers, etc.) One can only suppose that the modes in the present study can be described via some topological models, but this has not been done. Therefore, the word "topological" should be removed and used only in the discussion of *possible* interpretations or analogies.

2. It is worth mentioning that acoustic fields in a homogeneous medium can have only nonzero SAM *density*, while the integral SAM vanishes, in agreement with the spinless nature of phonons. This can be not so in inhomogeneous media, such as waveguides. The presence of nonzero integral SAM in the waveguide modes is quite nontrivial, I think.

3. In connection with the previous remark, I am wondering if the acoustic field has any tails outside the waveguide? Can these be described/measured to include into the consideration? Also, do the small rods of the metasurface scatter the guided mode to the outer space? Usually, any scatterer near the waveguide surface couples the guided mode to outside propagating modes (i.e., sound in the air).

4. It would be helpful to provide more details of experimental measurements. What are the materials, sources, and detectors used? What is the wavelength? How were the field and spin density measured inside the waveguide?

5. Since the studies of the acoustic SAM started only last year, and there are only few papers on this topic, it is worth citing two recent relevant papers:

Phys. Rev. B 99, 174310 (2019),

Phys. Rev. Lett. 123, 183901 (2019).

This would justify the claim "the acoustic SAM ... attracts considerable attention".

6. The authors use only the vector velocity field in their approach, ignoring the scalar pressure field. This approach is possible but not unique. In actual fact, the full "state vector" of the acoustic field has four components: (p,v) .

7. The authors explain the absence of backscattering by the fact that "the corner scattering cannot flip the acoustic spin". Why is this so? As far as I understand, the waveguide supports both the forward and backward modes with opposite spins. So, if there is coupling between these modes, there should be backscattering accompanied by the spin flip. This is what happens with the

backscattering of regular guided or surface modes by any impurity: the transverse spin is always there and it flips. Why cannot such a strong scatterer as the corner couple the opposite modes? I can only suppose that in the case under consideration the coupling between the opposite modes is small (but non-zero) due to their almost-orthogonal near-circular polarizations, and therefore backscattering is weak (but non-zero). The authors should elaborate on this and explain the phenomenon they observe.

8. The section about the geometric phase is misleading. The geometric phase appears from the spin-rotation coupling (analogous to the Coriolis effect), which is described by the scalar product of spin and angular velocity (as in the Hamiltonian mentioned in the text), see e.g. [20]. If the spin is orthogonal to the angular velocity, the geometric phase vanishes. This is exactly the case in this work. One can also notice that the geometric phase is linear in the angular velocity, while the phase considered in this paper is quadratic. As far as I understand, this is just a quadratic correction to the zero-order dynamical phase, and there is nothing geometric here. Therefore, this section should be removed from the paper.

9. In the Supplementary Materials, section I only repeats basic theoretical background from previously published works on this topic. Section II is misleading because there is no geometric phase. Therefore, these sections should be removed. Section III essentially contains only one figure without any explanations. What is it about? Is it a result of exact numerical simulations or just a schematic cartoon? How can the reader guess about the authors results and ideas from a figure without any explanations?

Reviewer #2 (Remarks to the Author):

Inspired by spin photonics which is a very hot field in recent years, there has been considerable interest in exploring analogous properties of sound systems. It has been reported that sound can also carry spin angular momentum (SAM) although it is a longitudinal wave. In this manuscript, the authors designed an acoustic waveguide that supports SAM and presented three applications: back-immune transport, spin-selective routing, and spin-redirection geometric phase effect. These applications could be of interest to researchers in the community of acoustic angular momentum. I suggest that the explanations of the first two applications should be improved to be more accurate, as the back-immunity/selective routing is not of topological origin. I consider that the third application is about a different phase other than the spin-redirection geometric phase. My detailed comments are as follows.

Major comments:

1. The backscattering-immune transport phenomena do not derive from a topological property of the waveguide. My understanding is that the discussed phenomena are derived from spatial symmetry under specific boundary condition. It is therefore not appropriate to refer to it as "topological transport" in the title and main text.
2. Regarding the results in Fig. 3, there is no physical mechanism that can prevent the flipping of spin at the corner since it breaks the mirror symmetry. The modification of the corner by the metasurface can improve this situation, but there should be certain reflection, although it might be weak. Again, this is because the transport is not protected by topology. Similar situation also exists for the T-shape corner.
3. The structure in Fig. 4 cannot give rise to spin-redirection geometric phase. The direction of rotation angular velocity Ω is perpendicular to the SAM, therefore, there is no $S \cdot \Omega$ term in the effective Hamiltonian. The spin-redirection geometric phase should be zero. This is also manifested by the expression of Φ_g in line 241, which depends on the wavevector k , while spin-redirection geometric phase only depends on SAM and the geometric parameters of the system.

The phase Φ_g could be a different phase induced by the rotation of metasurface structures.

Minor comments:

4. Line 83, "... which ensembles the relation of spin-momentum locking in topological states". What topological states of the waveguide do the authors refer to?

5. The authors should explain in the introduction part that why the existence of acoustic SAM does not contradict with the curl-free nature of longitudinal waves. The representation using either pressure field or velocity field does not affect this intrinsic property of sound as a longitudinal wave.

6. The generation of a π phase by the metasurface is a narrow-band phenomenon, which means the guided sound is in general "elliptically" polarized over a frequency range. It is better to show the variation of SAM within the considered frequency range [2850Hz,2950Hz]. If the authors have used a perfectly circular polarized dipole source, the mismatch between the source SAM and mode SAM could affect the results in Fig. 3.

7. Is the dispersion in Fig.1(c) numerical result? If yes, the author should provide axis labels.

Response to Reviewer 1:

This is an interesting paper, which reports an experimental observation of the robust acoustic-spin-controlled unidirectional transport in acoustic waveguides with metasurfaces. The presence of nonzero spin angular momentum density in structured sound waves was revealed very recently, and this direction of research was started by the authors of the current manuscript. Their present experimental demonstrations are novel and impressive. The topics of acoustic spin, metasurfaces, and unidirectional backscattering-immune transport are very hot, so this research is timely. However, I found a number of significant problems in this manuscript and interpretation of the results. Therefore, I will recommend publication of these results in Nature Communications only after the following questions are carefully considered and addressed by the authors, including some mandatory major changes in the manuscript.

Response:

Thanks for the reviewer's endorsements and suggestions that greatly improved our work. To our best efforts, we have work out a new revision and one-to-one response to address all the concerns and suggestions. Hope the reviewer will find our revised revision satisfactory.

1. The authors use the word "topological" in the title and throughout the paper. However, there is no topological theory supporting the phenomena observed here. There are only some symmetry-based arguments which involve the spin-direction locking in the modes of the metasurface-modified waveguide. (Note that the spin-momentum locking in electromagnetic/acoustic evanescent modes [3,15,20,24,29] has no clear topological origin, it should not be confused with truly topological phenomena involving bulk-boundary correspondence, topological numbers, etc.) One can only suppose that the modes in the present study can be described via some topological models, but this has not been done. Therefore, the word "topological" should be removed and used only in the discussion of *possible* interpretations or analogies.

Response:

Thanks for the reviewer's concerns about the word "topological". We have removed this word and re-written the relevant sentences in proper ways. As pointed out, our results were not described by some topological model or theory, but just with some symmetry-based arguments. The reviewer's concerns urged us to work out analytic theory analysis to clarify spin-momentum locking phenomena. We have added a symmetry discussion explaining the emergence of SAM, and the analytic result Eq. 2 explicitly showing the tight spin-momentum locking, with an additional Method section of theory details. Thanks to the reviewer's suggestion, those added contents significantly improve our work and make conclusions more rigorous than before.

Added details are in the Method section of the revised main text, as follows:

In the cylindrical coordination system (r, φ, z) as shown in Fig. A1 (below), the solution

of the acoustic field in the metasurface waveguide can be represented as the linear composition of the Bessel function $J_\ell(r)$ as: $p = p_0 \sum_\ell c_\ell J_\ell(\kappa r) e^{i\ell\varphi} e^{i(kz-\omega t)}$, where c_ℓ is the modal amplitude, $\kappa = \sqrt{\frac{\omega^2}{c^2} - k^2}$, k is the wave vector, ω is the frequency and c is the sound speed in the air. Due to the non-symmetric boundary conditions, we will have: $p|_{r=R} = 0$ for $\varphi \in [-\frac{\pi}{2}, \frac{\pi}{2}]$ and $\partial_r p|_{r=R} = 0$ for $\varphi \in [-\pi, -\frac{\pi}{2}) \cup (\frac{\pi}{2}, \pi]$, where R is the radius of the waveguide. For solving with the ground state of the metasurface waveguide, only $\ell = 0, \pm 1$ are considered. After some mathematical derivations considering the special boundary condition, the acoustic field for the ground state in the metasurface waveguide can be approximately represented as:

$$p = p_0 (J_0(\kappa r) + \alpha J_1(\kappa r) \cos(\varphi)) e^{i(kz-\omega t)}$$

with the dispersion $k^2 + \kappa^2 = \frac{\omega^2}{c^2}$, α is a constant. With the boundary conditions, we will have that $\kappa R = \xi$, $\alpha = -\frac{J_0(\xi)}{J_1(\xi)}$, and ξ is the first root of the equation $J_1^2(\xi) - \frac{1}{2}J_0(\xi)(J_0(\xi) - J_2(\xi)) = 0$ and $\xi > 0$.

Fig. A1. the polar coordination in the metasurface waveguide. The gray and yellow color denote the conventional sound hard boundary and metasurface boundary respectively. Here, we consider the reflection phase of the metasurface is $\phi = \pi$ (corresponding to the sound soft boundary).

The velocity field $\mathbf{v} = -\frac{i}{\rho\omega} \nabla p$ will be:

$$v_r = \frac{p_0}{\rho\omega} \frac{i\kappa}{2} (2J_1(\kappa r) + \alpha(J_2(\kappa r) - J_0(\kappa r)) \cos(\varphi)) e^{i(kz-\omega t)}$$

$$v_\varphi = \frac{p_0}{\rho\omega} \frac{i\alpha}{r} J_1(\kappa r) \sin(\varphi) e^{i(kz-\omega t)}$$

$$v_z = \frac{p_0}{\rho\omega} k (J_0(\kappa r) + \alpha J_1(\kappa r) \cos(\varphi)) e^{i(kz-\omega t)}$$

and the corresponding acoustic SAM density $\mathbf{s} = \frac{\rho}{2\omega} \text{Im}[\mathbf{v}^* \times \mathbf{v}]$ will be:

$$s_r = -\frac{|p_0|^2}{\rho\omega^3} \frac{k\alpha}{r} J_1(\kappa r) (J_0(\kappa r) + \alpha J_1(\kappa r) \cos(\varphi)) \sin(\varphi)$$

$$s_\varphi = -\frac{|p_0|^2}{\rho\omega^3} \frac{k\kappa}{2} (J_0(\kappa r) + \alpha J_1(\kappa r) \cos(\varphi)) (-2J_1(\kappa r) + \alpha(J_0(\kappa r) - J_2(\kappa r)) \cos(\varphi))$$

$$s_z = 0$$

The s_x and s_y will be: $s_x = s_r \cos(\varphi) - s_\varphi \sin(\varphi)$, $s_y = s_r \sin(\varphi) + s_\varphi \cos(\varphi)$. We have plotted the theoretical results, numerical simulations and experimental measurements in Fig.A2 (the Fig.3 of main text). We can see that the theoretical results and numerical simulations are in good agreements with experimental measurements.

Fig. A2. The comparison between the theoretical results, numerical simulations and experimental measurements for spin angular momentum density in the metasurface waveguide mode. The experimental detection frequency is $f = 2.9\text{kHz}$.

The total SAM will be integral of acoustic SAM density as $\mathbf{S} = \int \mathbf{s} d\mathbf{r}^2$ and

$$S_x = \int_0^R \int_0^{2\pi} s_x r dr d\varphi = 0$$

$$S_y = \int_0^R \int_0^{2\pi} s_y r dr d\varphi = \frac{\pi |p_0|^2}{\rho \omega^3} k R J_0^2(\kappa R)$$

$$S_z = \int_0^R \int_0^{2\pi} s_z r dr d\varphi = 0$$

So, the total SAM for the ground state of metasurface waveguide will be:

$$\mathbf{S} = \frac{\pi |p_0|^2}{\rho \omega^3} k R J_0^2(\kappa R) \mathbf{e}_y$$

It is clear that the metasurface waveguide modes will naturally carry non-zero SAM $S_y \neq 0$ and importantly, this SAM will strongly depend on the momentum direction k , which resembles the spin-momentum locking in QSHE. This spin-momentum locking in waveguide bulk modes is the key point of our work.

2. It is worth mentioning that acoustic fields in a homogeneous medium can have only nonzero SAM *density*, while the integral SAM vanishes, in agreement with the spinless nature of phonons. This can be not so in inhomogeneous media, such as waveguides. The presence of nonzero integral SAM in the waveguide modes is quite nontrivial, I think.

Response:

Thanks for the reviewer's endorsement about the non-zero SAM in metasurface

waveguides. We really much appreciate.

3. In connection with the previous remark, I am wondering if the acoustic field has any tails outside the waveguide? Can these be described/measured to include into the consideration? Also, do the small rods of the metasurface scatter the guided mode to the outer space? Usually, any scatterer near the waveguide surface couples the guided mode to outside propagating modes (i.e., sound in the air).

Response:

Thanks for the reviewer's interests about the waveguide settings. The tube surface is physically a sound hard boundary so that there are no tails outside waveguide. Because all hard/metasurface boundaries are set as all reflection and no transmission. Only their reflected phases are different. All the small rods of the metasurface are also set as hard boundary conditions. Those settings follow the convention of acoustic study and the experiment results are consistent with exact simulations [please also refer to the newly added Fig. 3].

For experiments, the metasurface waveguide is constructed by 3D print resin material with 2mm thickness, which can reflect the acoustic wave in the nearly same amplitude with the incident wave due to the big difference on mass densities between the 3D print media and the air. These similar material considerations have been involved in many other acoustic works [*Nature* 560.7716 (2018): 61-64; *Nature Physics* 13.4 (2017): 369-374; *Nature Physics* 14.1 (2018): 30-34] and proved to be in good agreements with both theoretical and numerical predictions. Sometimes, may due to fabrication flaws or effect surface depth, there would be some leaky modes in actual experiments. However, the main phenomena predicted theoretically and numerically are observed experimentally, proving that these possible leaky modes are weak.

4. It would be helpful to provide more details of experimental measurements. What are the materials, sources, and detectors used? What is the wavelength? How were the field and spin density measured inside the waveguide?

Response:

Thanks for the reviewer's suggestion. We have added more experimental details into the section Method of the revised main text, as "Sample fabrication and experimental measurements."

All the elements are fabricated using 3D printing with the photosensitive resin material (modulus 2765 MPa, density 1.3 g/cm³), and can be treated as acoustically rigid. The elements were 3D-printed, one unit cell at a time, and were connected to form different configurations. The 1/4-inch microphone (GRAS Type 46BE) is placed at the measuring terminal of the sample to detect amplitude and phase of the sound. The experimental environment was an open space to avoid unnecessary reflection. The detected signals are acquired by the NI 9234 data acquisition module. A sine signal is driven by the waveform generator (Keysight 33500B) with frequency range from 2.7 kHz to 3.2 kHz. The sine signal was fed into a loudspeaker as the sound source. The sound waves emitted by the loudspeaker are guided into the tube. The position of the microphone is controlled by the automatic stage.

Fig. A3. The experimental scheme to detect the SAM density. The acoustic detector is inserted into the waveguide, moved by the autostage and recording the acoustic field.

The acoustic field inside the waveguide is measured by placing the acoustic detector into the waveguide (the scale of the microphone is much smaller than the radius of the cross section of the waveguide), moving the acoustic detector with the automatic stage and recording the acoustic pressure field p_{exp} , shown in Fig.A3. The experimental SAM densities is obtained by $s_{exp} = \frac{\rho}{2\omega} \text{Im}[\mathbf{v}^* \times \mathbf{v}]$ and $\mathbf{v}_{exp} = -\frac{i}{\rho\omega} \nabla p_{exp}$. We have added the details of the approach for measuring field inside the waveguide in the main text.

The wave vector k inside the waveguide is measured by moving the acoustic detector along the z axis gradually and recording the phase change during the relocation. The wave vector k will be the ratio between the phase change and the position change along z axis. For 2.85kHz and 2.9 kHz in the phase detection section, the measured wavelength of the waveguide mode is about 24.5°/cm.

5. Since the studies of the acoustic SAM started only last year, and there are only few papers on this topic, it is worth citing two recent relevant papers:

Phys. Rev. B 99, 174310 (2019),

Phys. Rev. Lett. 123, 183901 (2019).

This would justify the claim “the acoustic SAM … attracts considerable attention”.

Response:

Thanks for the reviewer’s supply of these relevant papers. Based on the reviewer’s suggestions, we have cited these references as Ref.[27,28] in introduction part, as: “*exploiting the acoustic SAM in practice becomes of significant importance and attracts considerable attentions, including spin-orbit coupling in Bessel beams [27], acoustic spin-induced torque [28].*”

6. The authors use only the vector velocity field in their approach, ignoring the scalar pressure field. This approach is possible but not unique. In actual fact, the full “state vector” of the acoustic field has four components: (p, \mathbf{v}) .

Response:

Thanks for the reviewer’s comments. We agree with the opinion that the full “state vector” of the acoustic field is more proper to contain four components (p, \mathbf{v}) . But presently the

acoustic SAM only associates with the velocity field \mathbf{v} , while the p does not contribute. As such, velocity field alone is enough to describe the acoustic SAM. Therefore, to avoid any confusions, we have removed so that avoided the usages of “state vector of acoustic field”.

7. The authors explain the absence of backscattering by the fact that “the corner scattering cannot flip the acoustic spin”. Why is this so? As far as I understand, the waveguide supports both the forward and backward modes with opposite spins. So, if there is coupling between these modes, there should be backscattering accompanied by the spin flip. This is what happens with the backscattering of regular guided or surface modes by any impurity: the transverse spin is always there and it flips. Why cannot such a strong scatterer as the corner couple the opposite modes? I can only suppose that in the case under consideration the coupling between the opposite modes is small (but non-zero) due to their almost-orthogonal near-circular polarizations, and therefore backscattering is weak (but non-zero). The authors should elaborate on this and explain the phenomenon they observe.

Response:

Thanks for the reviewer’s concerns about the suppression of corner-scattering. We agree with the reviewer that the backscattering is non-zero, but weak. We also agree with the reviewer’s insightful comments.

Firstly, in the metasurface waveguide, the reflected wave carries not only different wave vector, but also the opposite acoustic SAM. Acoustic spin texture represents the near-circular polarization velocity fields (as demonstrated in Fig.A4). As the consequence of the non-zero SAM, this momentum-dependent polarization will lead to a low coupling between opposite propagating modes. To describe this forward-backward mode coupling, we calculate $\eta = \frac{\int_S |\mathcal{F}_2^* \cdot \mathcal{F}_1|^2 dr^2}{\int_S |\mathcal{F}_1^* \cdot \mathcal{F}_1|^2 dr^2}$, as the overlap (coupling) between two opposite modes, where $\mathcal{F} = \left(v_x, v_y, v_z, -\frac{i}{Z_0} p \right)^T$ describes the acoustic field, $Z_0 = \rho c$ is the characteristic specific acoustic impedance of air, \mathcal{F}_1 and \mathcal{F}_2 denote the forward ($k > 0$) and reverse ($k < 0$) modes at the same frequency respectively, and S means the cross-section of waveguide. According to analytical expressions of velocity and pressure in our newly added theory, the values will be $\eta = 4.2\% \sim 3.6\%$ in the frequency region [2.85, 2.95]kHz. For numerical simulations, we find this value is $\eta \approx 6\%$ at 2.9 kHz. The low value of η has proven that the approximately orthogonal relation between forward and reverse modes. Similarly, in the T-shape spin-selective routing experiment, this approximately orthogonal relation also explains that the scattering fields from the corner will have low couplings to fields (direction) with opposite spin.

Secondly, the scattering behaviors of scatters highly depend on the type of scatter and its interaction with wave modes. For example, some special impurities and defects (breaking time-reversal symmetry) indeed will break topological protections in topological insulators [*PNAS* 113.18 (2016): 4924-4928; *Nature communications* 7.1 (2016): 1-9.]. One can find spin-related scatters in many polarization dependent metasurface research works [*Nanophotonics* 6.1 (2016): 215-234; *Nanophotonics* 6.1 (2017): 51; *Science* 340.6133 (2013): 724-726]. Therefore, when a scatter is not strong enough and preserve the time-reversal symmetry (i.e., SAM conservation),

the scatter does not have the ability to flip the acoustic spin texture to ensure SAM matching. As such, the scatter cannot flip the acoustic SAM to match the modes so that the backscattering would be suppressed due to the mismatch of polarization profiles. According to our experiments, these metasurface waveguide modes is insensitive to corners.

Thus, although facing such a conventionally “strong” corner scatter, the metasurface waveguide modes will pass through the corner with low backscattering. We have added more discussions in the revision, which are highlighted in blue color.

Fig. A4. The generally elliptically (near-circularly) polarized velocity fields of modes simulated at the center of the waveguide for $k > 0$ in the frequency region [2.85, 2.95] kHz. The purple arrow and the arrowed red circle denote the polarized velocity field and its time evolution. The plotted velocity field is normalized by the amplitude of v_z . The modes for $k < 0$ will have opposite polarized fields due to the time-reversal symmetry.

8. The section about the geometric phase is misleading. The geometric phase appears from the spin-rotation coupling (analogous to the Coriolis effect), which is described by the scalar product of spin and angular velocity (as in the Hamiltonian mentioned in the text), see e.g. [20]. If the spin is orthogonal to the angular velocity, the geometric phase vanishes. This is exactly the case in this work. One can also notice that the geometric phase is linear in the angular velocity, while the phase considered in this paper is quadratic. As far as I understand, this is just a quadratic correction to the zero-order dynamical phase, and there is nothing geometric here. Therefore, this section should be removed from the paper.

Response:

Thanks for the reviewer’s insightful comments about the old section of geometrical phase. Based on the reviewer’s mentions, we have re-examined our system settings and theoretical descriptions carefully. We agree with the reviewer that the acoustic spin in waveguides is geometrically orthogonal with the angular velocity. In this case, the geometrical phase will vanish. Thus, we re-performed the theoretical analysis to explain this phase. We hope the referee will agree that we have found a reasonable compromise in the revised version of our manuscript. We keep this part of rotating metasurface because we think it shows interesting features of rotated acoustic spin and a phase modulator.

In the main text, an effective magnetic field for SAM is induced by rotating the metasurface waveguides and causes the effective spin precession: $\frac{\partial}{\partial z} \mathbf{S} = \mathbf{\Omega} \times \mathbf{S}$. The $\mathbf{\Omega}$ can be the effective magnetic field for the spin in the waveguide and also the Larmor frequency for this precession. Due to this spin precession, the center position \mathbf{R}_0 of the waveguide mode will rotate along z axis following $\frac{\partial}{\partial z} \mathbf{R}_0 = \mathbf{\Omega} \times \mathbf{R}_0$ in Fig.A5(b), which induces the additional phase. This is because the center of the waveguide mode mainly located off-center at $R_0 = \sqrt{x_0^2 + y_0^2}$, as shown in Fig.A5(a), where $x_0 = \frac{\int_{\mathcal{S}} x |p|^2 dr^2}{\int_{\mathcal{S}} |p|^2 dr^2} = -0.014m$, $y_0 = \frac{\int_{\mathcal{S}} y |p|^2 dr^2}{\int_{\mathcal{S}} |p|^2 dr^2} = 0m$ and \mathcal{S} denotes the cross-section of the metasurface waveguide. After we rotate the metasurface waveguide namely $\mathbf{\Omega} \neq 0$, the effective propagation length will increase in Fig.A5(b). The effective propagation length will be $\sqrt{L^2 + (\theta R_0)^2}$ in Fig. A5(c), where $\theta = \Omega L$. Thus, the phase Θ accumulated during rotating metasurface waveguide when $\mathbf{\Omega} \neq 0$ will be $\Theta = k\sqrt{L^2 + (\theta R_0)^2} \approx kL + \frac{kR_0^2}{2} \Omega^2 L$. We can see that in addition to the unperturbed phase $\Theta_0 = kL$, the addition phase will be $\gamma = \frac{kR_0^2}{2} \Omega^2 L = \frac{kR_0^2}{2L} \theta^2$.

Fig. A5. (a) the center position (Red point) of the metasurface waveguide mode with a distance R_0 off-center. (b) the effective propagation path (Red line) will increase after rotating boundary conditions, namely $\mathbf{\Omega} \neq 0$. Gray line denotes the center of the waveguide. (c) the effective propagation path can be represented by the length of the waveguide L and the arc length $R_0\theta$, namely $\sqrt{L^2 + R_0^2\theta^2}$.

Although the discussed phase is not the geometrical phase originated from spin-rotation coupling, these phenomena (i.e, rotated spin textures and phase accumulation) will improve us the understanding about the SAM in the metasurface waveguide. We keep this part of rotating metasurface because we think it shows interesting features of rotated acoustic spin and a phase

modulator. We hope the referee will agree that we have found a reasonable compromise in the revised version of our manuscript. Finally, we deeply appreciate the reviewer's deep insights about this part, which really improves our manuscript significantly.

9. In the Supplementary Materials, section I only repeats basic theoretical background from previously published works on this topic. Section II is misleading because there is no geometric phase. Therefore, these sections should be removed. Section III essentially contains only one figure without any explanations. What is it about? Is it a result of exact numerical simulations or just a schematic cartoon? How can the reader guess about the authors results and ideas from a figure without any explanations?

Response:

We have followed the reviewer's suggestion and removed these two Sections (I,II). We are sorry about insufficient descriptions in the Section III. We have added more descriptions and discussions for the figure that is a result of numerical simulations. The content of this section III is to show extra cases about the robustness of the acoustic transport in metasurface waveguides.

Response to Reviewer 2:

Inspired by spin photonics which is a very hot field in recent years, there has been considerable interest in exploring analogous properties of sound systems. It has been reported that sound can also carry spin angular momentum (SAM) although it is a longitudinal wave. In this manuscript, the authors designed an acoustic waveguide that supports SAM and presented three applications: back-immune transport, spin-selective routing, and spin-redirection geometric phase effect. These applications could be of interest to researchers in the community of acoustic angular momentum. I suggest that the explanations of the first two applications should be improved to be more accurate, as the back-immunity/selective routing is not of topological origin. I consider that the third application is about a different phase other than the spin-redirection geometric phase. My detailed comments are as follows.

Response:

Thanks for the reviewer's endorsements, and suggestions about our work. According to the suggestion, we have worked out a theory to analytically obtain the velocities and pressure in the metasurface waveguide, which finally provides us the more accurate explanation of acoustic spin-momentum locking [please refer to Eq. 2 and added method for details]. We also correct the discussion of the last part. We thank the reviewer for his constructive suggestions that greatly improved our work. Hope the reviewer will find our revised revision satisfactory.

Major comments:

1. The backscattering-immune transport phenomena do not derive from a topological property of the waveguide. My understanding is that the discussed phenomena are derived from spatial symmetry under specific boundary condition. It is therefore not appropriate to refer to it as “topological transport” in the title and main text.

Response:

Thanks for the reviewer’s concerns about the word “topological”. We have removed this word and re-written the relative sentences in proper ways. As pointed out, our presented work is mainly based on mirror symmetry breaking under non-symmetric boundary conditions. In the previous version, the main reason that we call it “topological” is that the presented metasurface modes have tight spin-momentum locking resembling the main feature of the quantum spin Hall effect. The key points of our work are spin-momentum locking of metasurface waveguide bulk modes and several spin-related transport phenomena.

We have significantly improved our work and added some theoretical details about spin-momentum locking, as highlighted in the main text of revision [see also the reply to Reviewer 1’s Q1].

2. Regarding the results in Fig. 3, there is no physical mechanism that can prevent the flipping of spin at the corner since it breaks the mirror symmetry. The modification of the corner by the metasurface can improve this situation, but there should be certain reflection, although it might be weak. Again, this is because the transport is not protected by topology. Similar situation also exists for the T-shape corner.

Response:

Thanks for the reviewer’s concerns about the physical mechanism of scattering suppressions. Based on the reviewer’s concerns, we have added the discussions about these parts. Admittedly, these transports are not protected by topological models and there will be some certain but weak reflections when passing through corners. What we want to demonstrate is that introducing the spin-momentum locking for waveguide bulk modes will be helpful to suppress these scatterings compared with conventional “spinless” duct waveguide modes. As pointed out by Reviewer 1 in his Q7 “*in the case under consideration the coupling between the opposite modes is small (but non-zero) due to their almost-orthogonal near-circular polarizations, and therefore backscattering is weak (but non-zero)*.”

Please see the details analysis in the response to Reviewer1’s Q7. Thanks very for the concern, which helped us to further clarify the results.

3. The structure in Fig. 4 cannot give rise to spin-redirected geometric phase. The direction of rotation angular velocity Ω is perpendicular to the SAM, therefore, there is no $\mathbf{S} \cdot \Omega$ term in the effective Hamiltonian. The spin-redirected geometric phase should be zero. This is also manifested by the expression of Φ_g in line 241, which depends on the wavevector \mathbf{k} , while spin-redirected geometric phase only depends on SAM and the geometric parameters of the system. The phase Φ_g could be a different phase induced by the rotation of metasurface structures.

Response:

Thanks for the reviewer's insightful comments about the geometrical phase. We agree with the reviewer. Thus, we remove the geometric phase explanation, and re-perform the theoretical analysis to explain this phase with a new one. Please refer to the reply to Reviewer 1's Q8.

Although the discussed phase is not the geometrical phase originated from spin-rotation coupling, these phenomena (i.e, rotated spin textures and phase accumulation) will improve us the understanding about the SAM in the metasurface waveguide. We keep this part of rotating metasurface because we think it shows interesting features of rotated acoustic spin and a phase modulator. We much appreciate the reviewer's deep insights about this part, which really improves our manuscript significantly.

Minor comments:

4. Line 83, "..., which ensembles the relation of spin-momentum locking in topological states". What topological states of the waveguide do the authors refer to?

Response:

Thanks for the reviewer's concerns about some statements. We are sorry about some typos here: "ensemble" should be "resemble". The mentioned topological states are quantum spin Hall effect states or surface states on the topological insulators. The main feature of edge-states in quantum spin Hall effect is spin-momentum locking due to spin-orbit coupling. Based on the reviewer's comments, we have re-written it as "resembling the relation of spin-momentum locking in QSHE states."

5. The authors should explain in the introduction part that why the existence of acoustic SAM does not contradict with the curl-free nature of longitudinal waves. The representation using either pressure field or velocity field does not affect this intrinsic property of sound as a longitudinal wave.

Response:

Thanks for the reviewers' comments. We have added discussions about this part into the introduction, highlighted as blue color.

On the one hand, most people thought that the longitudinal wave cannot possess SAM due to its lack of abilities ($\nabla \times \mathbf{v} = 0$) to form circularly polarized plane wave like transverse optical waves ($\nabla \times \mathbf{v} \neq 0$). A naive thought then came out that the curl-free nature of longitudinal waves forbids the existence of acoustic SAM, because $\nabla \times \mathbf{v} = 0$ indicates the zero vorticity. These thoughts restrict the development of the concept of acoustic SAM. However, zero vorticity $\nabla \times \mathbf{v} = 0$ does not mean that $\text{Im}[\mathbf{v}^* \times \mathbf{v}] = 0$, so that the existence of acoustic SAM does not contradict with the curl-free nature of longitudinal waves.

On the other hand, conventionally it is enough to use only the scalar pressure field to describe the acoustic wave in many ways, because even the curl-free longitudinal velocity fields can be described by the gradient of scalar pressure field. While in classical field theory [see the textbook by Soper, D. E. "*Classical field theory*"], it has been proved that the scalar field is spinless. So, people believed in a long time that the curl-free longitudinal acoustic wave does

not have SAM. However, as pointed in recent publications that acoustic field is a vectorial field that we need take into account the velocities in full. This is also pointed by Reviewer 1 that “*the full “state vector” of the acoustic field has four components: (p, v).*” Therefore, there is no conflict in principle.

6. The generation of a Pi phase by the metasurface is a narrow-band phenomenon, which means the guided sound is in general “elliptically” polarized over a frequency range. It is better to show the variation of SAM within the considered frequency range [2850Hz,2950Hz]. If the authors have used a perfectly circular polarized dipole source, the mismatch between the source SAM and mode SAM could affect the results in Fig. 3.

Response.

Thanks for the reviewer’s insightful suggestions. We have added these polarization results into the main text and Supplementary. Yes, the sound in metasurface waveguide will be in general “elliptically” polarized over the frequency range [2.85, 2.95] kHz. Because the SAM is directly related to the near-circularly (generally elliptically) polarized velocity field, we obtain polarization forms of velocity fields in the center of the waveguide for different frequencies in Fig.B1 in below. Admittedly, it will make influences to the excitation efficiency due to the mismatch between sources and local polarized profiles in Fig. 1. But this is not the key point of our experimental work.

In the experimental measurements of Fig.3 [see also the schematic Fig. 2d and the Method about experimental details], the source is a simple loudspeaker, which can be regarded as the incident plane wave but not circularly polarized dipoles. The mismatch between the source and the mode of SAM in the waveguide will affect the efficiency of the incidence. But once the sound is injected from the loudspeaker into the waveguide, the acoustic field will automatically follow the tube’s eigenmode to form the chiral polarized profile of nonzero SAM.

Although the SAM of the waveguide eigenmode may lead some changes when frequency changes, according to the experimental results in Fig.3, this influence from this SAM variation seems to be small. Our experimental results, numerical simulations, and analytical theory are consistent with each other quite well.

Fig B1. The generally elliptically (near-circularly) polarized velocity fields of modes simulated at the center of the waveguide for $k > 0$ in the frequency region [2.85, 2.95] kHz. The purple arrow and the arrowed red circle denote the polarized velocity field and its time evolution. The plotted velocity field is normalized by the amplitude of v_z . The modes for $k < 0$ will have opposite polarized fields due to the

time-reversal symmetry.

7. Is the dispersion in Fig.1(c) numerical result? If yes, the author should provide axis labels.

Response:

Thanks for the reviewer's mentions. We have added the axis labels.

Main Change List

1. Removed the inappropriate usages of the word “topological” and/or replaced it with proper descriptions (suggested by both reviewers);
2. Added more descriptions in the introduction (for the concern of Reviewer 2);
3. Added theoretical analysis (especially the analytic Eq. 2) for spin-momentum locking (follow the suggestion of Reviewer 1), with an additional Method section for details;
4. Added axis labels for Fig.1(c) of the main text;
5. Re-adjusted the Fig.2 and add a new Fig.S1 to demonstrate frequency (in)-dependence of elliptical polarized velocity fields (for the concern of Reviewer 2);
6. Added a new Fig.3 to confirm that theoretical, numerical and experimental results about SAM densities are consistent with each other quite well (to respond to both reviewers);
7. Added details about experimental detections (for the concern of Reviewer 1);
8. Added more discussions about physical mechanisms of the chiral velocity fields and robustness of the acoustic spin transport; (to respond to both reviewers)
9. Corrected the previous inaccurate statements about geometric phase (suggested by both reviewers);
10. Removed the previous Section I, II of Supplementary (suggested by Reviewer 1);
11. Added details about simulation cases in Section II of Supplementary.

All main changes are highlighted in Blue color in the revised main text.

REVIEWER COMMENTS

Reviewer #1 (Remarks to the Author):

I think the authors have considerably improved the manuscript and addressed most of the issues raised in the referees' reports. Therefore, I recommend to proceed with the publication of this manuscript in Nature Communications.

Two issues to be considered by the authors before the final acceptance:

1. It seems that the new equation (2) should indicate the transverse spin-momentum locking via the spin proportionality to k . If so, this should be made mathematically consistent. Namely, the authors write "k is the wave vector". This is not true, k is obviously a scalar, not a vector. Perhaps k is the wave number? Then, this is a purely positive number, which cannot reflect the spin-momentum locking. Probably, there should be the longitudinal momentum (wave vector) component $k=k_z$, which can be both positive and negative and indeed indicates the spin-momentum locking. This should be described accurately.
2. From the response letter, it seems that the authors think that spin can be flipped only by scatterers with time-reversal symmetry breaking. This is not so in optics/acoustics. Optical spin is not conserved in reflections and scatterings preserving time-reversal symmetry. As far as I understand, the only thing that suppresses the back-reflection from corners/inhomogeneities is the near-zero overlap (coupling) coefficient η between the forward and backward modes. But no symmetry arguments forbid backscattering and spin flip.

Reviewer #2 (Remarks to the Author):

The authors addressed my comments very carefully. Overall, the manuscript presents an interesting work. However, I still have concerns about certain parts of the revisions made by the authors. I cannot recommend it for publication unless the authors can completely address my comments below:

1. The symmetry argument newly added to the first graph of Page 2 is misleading. It does not justify the requirement of x-mirror-symmetry breaking of the waveguide geometry. The presence of a x-mirror symmetry (M_x) only indicates that the waveguide supports two opposite spins (S_y and $-S_y$), however, it does not necessarily mean S_y must be vanished. To understand this point, consider light in free space, a mirror operation on LCP light can turn it into RCP light, but it does not indicate that, to excite a LCP light, one has to break the corresponding mirror symmetry of free space.
2. The authors added an analytical theory to support the existence of non-zero total spin and its k -dependence. While this is a good try and it certainly helps to understand the phenomena, the theory is unfortunately flawed. Decomposing the waveguide fields using Bessel function basis only works in systems with cylindrical symmetry where Bessel functions can form complete and orthogonal basis. The requirement of cylindrical symmetry allows separating the eigen wavefields into product of an azimuthal function and a radial function by separation of variables. The waveguide here apparently does not belong to these cases since the boundary decoration cannot be considered as a small perturbation to ideal cylindrical waveguide. This could be the reason that the analytical theory gives incorrect spin values near the boundary in Fig. 3.
3. In addition to the above comment, Eq. (3) of the analytical theory in the "Method" section is also problematic. The pressure field is decomposed into monopole and dipole modes. However, the

wavevectors of the two modes were taken to be equal, while it is known that they have different transverse and longitudinal components of wavevectors in general.

4. As also commented by the other referee, it is important to explain that why the waveguide corners do not induce strong reflections. In the revised manuscript, the authors only explain on Page 4 that this is because the corners “preserves SAM and fail to induce strong backscattering” and “backscattering requires strong scatters that reverse spin”. Why do the “strong” corners preserve spin? The pair of spin states are related by mirror operation and time-reversal operation, i.e., any scatterer applies a mirror operation (with mirror plane perpendicular to x or z axis) or time-reversal operation to one spin will transform it into the opposite spin and, therefore, cause reflections. Both the U-shape corner and the T-shape corner apply z-mirror operation on the spin state and, therefore, must induce certain reflection. The reflections in the proposed systems are weak. This might be attributed to the asymmetric distribution of the waveguide field, i.e., the pressure field mainly localize on one half of the waveguide cross section without cylinder decorations. The asymmetry of the waveguide field render it easily turn left than right in the U-shape case, and easily coupled to the left part than the right part in the T-shape case. If the U-shape waveguide is bended towards the decorated boundary, the reflection should be much stronger. Strong reflection can also happen in the T-shape case if the boundary decoration of the horizontal part faces upward. Therefore, it is fine to state that spin-momentum locking can assist in asymmetric coupling at the corners, but the weak reflection is not due to the near-orthogonality of the two spin states.

5. While the spin rotation part is a bit trivial, it can be kept in the main text as a demonstration of spin manipulation phenomenon.

Main change list

1. Page 2: Modify the statements about k of Eq. (2) (suggested by the referee 1);
2. Page 2: Improve the statements about mirror symmetry arguments (for the concern 1 of the referee 2);
3. Page 3,4: Rewrite some statements for clarifying mechanism and properties of weak reflections and scattering suppressions (for the concern 4 of the referee 2);
4. Page 4: Add the Fig.3 to clarify the transmission properties of the metasurface waveguide (for the concern 4 of the referee 2);
5. Page 7: Clarify the reasons of inaccurate spin near boundaries (for the concern 2 of the referee 2);

All main changes have been denoted in the blue color.

Response to Reviewer #1:

I think the authors have considerably improved the manuscript and addressed most of the issues raised in the referees' reports. Therefore, I recommend to proceed with the publication of this manuscript in Nature Communications.

Response:

Thanks very much for the reviewer's recommendations and efforts during this difficult time due to COVID-19.

Two issues to be considered by the authors before the final acceptance:

1. It seems that the new equation (2) should indicate the transverse spin-momentum locking via the spin proportionality to k . If so, this should be made mathematically consistent. Namely, the authors write " k is the wave vector". This is not true, k is obviously a scalar, not a vector. Perhaps k is the wave number? Then, this is a purely positive number, which cannot reflect the spin-momentum locking. Probably, there should be the longitudinal momentum (wave vector) component $k=k_z$, which can be both positive and negative and indeed indicates the spin-momentum locking. This should be described accurately.

Response:

We thank the referee for pointing out this. Yes, k is indeed "*the longitudinal momentum (wave vector) component $k=k_z$, which can be both positive and negative and indeed indicates the spin-momentum locking.*" We have improved the description based on the referee's suggestion.

2. From the response letter, it seems that the authors think that spin can be flipped only by scatterers with time-reversal symmetry breaking. This is not so in optics/acoustics. Optical spin is not conserved in reflections and scatterings preserving time-reversal symmetry. As far as I understand, the only thing that suppresses the back-reflection from corners/inhomogeneities is the near-zero overlap (coupling) coefficient η between the forward and backward modes. But no symmetry arguments forbid backscattering and spin flip.

Response:

Thanks for the referee's interests about the mechanism for suppressing back-reflections. We agree with the referee that, the suppression of acoustic spin flip (back-scattering) is NOT due to the absence of time-reversal symmetry (or mirror symmetry) breaking, but assisted by "*the near-zero coupling coefficient η between the forward and backward modes*" "*suppresses the back-reflection*" in systems. We mainly apply the symmetry arguments for explaining the physical origin of non-zero SAMs. "No symmetry arguments forbid backscattering and spin flip" in our case.

We have further modified and improved our presentation and language, to avoid any confusion.

Response to Reviewer #2:

The authors addressed my comments very carefully. Overall, the manuscript presents an interesting work. However, I still have concerns about certain parts of the revisions made by the authors. I cannot recommend it for publication unless the authors can completely address my comments below:

Response:

Thank the referee very much for the review efforts during this difficult time due to COVID-19.

1. The symmetry argument newly added to the first graph of Page 2 is misleading. It does not justify the requirement of x-mirror-symmetry breaking of the waveguide geometry. The presence of a x-mirror symmetry (M_x) only indicates that the waveguide supports two opposite spins (S_y and $-S_y$), however, it does not necessarily mean S_y must be vanished. To understand this point, consider light in free space, a mirror operation on LCP light can turn it into RCP light, but it does not indicate that, to excite a LCP light, one has to break the corresponding mirror symmetry of free space.

Response:

Thank the referee for pointing out this unclear point. We have modified the presentation to improve the symmetry argument. The important condition for our previous argument is that only **nondegenerate** mode exists for the waveguide eigenmode. In this nondegenerate condition, the single mode is x-mirror symmetric, indicates $s_y(x) = \mathcal{M}_x s_y(x) \mathcal{M}_x^{-1} = -s_y(-x)$, so that the nondegenerate mode will only carry the zero SAM $S_y = \int s_y dr^2$ due to the SAM cancellation. After introducing x-mirror symmetry breaking for this nondegenerate mode, there would be the non-zero SAM due to the non-perfect SAM cancellation.

We have added more detailed explanation to clarify this point.

2. The authors added an analytical theory to support the existence of non-zero total spin and its k-dependence. While this is a good try and it certainly helps to understand the phenomena, the theory is unfortunately flawed. Decomposing the waveguide fields using Bessel function basis only works in systems with cylindrical symmetry where Bessel functions can form complete and orthogonal basis. The requirement of cylindrical symmetry allows separating the eigen wavefields into product of an azimuthal function and a radial function by separation of variables. The waveguide here apparently does not belong to these cases since the

boundary decoration cannot be considered as a small perturbation to ideal cylindrical waveguide. This could be the reason that the analytical theory gives incorrect spin values near the boundary in Fig. 3.

Response:

We thank the referee for concerns about the approximated theoretical analysis. Please allow us to explain in details: The Bessel functions are the solutions for wave equations in the **cylindrical coordination**, which however does not mean that Bessel functions only works with **cylindrical symmetry**. In fact, the Bessel functions have been widely applied to solve eigenmodes in systems **without cylindrical symmetry**, e.g., semicircular waveguides, wedge-shape circular waveguides or sectoral waveguide. Some details could be found in textbooks and some published papers [Cheng D K. *Field and wave electromagnetics*. 1989; *IEEE Antennas and Propagation Magazine*, 1991, 33(6): 20-27; *Journal of Microwave Power and Electromagnetic Energy*, 2019, 53(4): 276-295.]

As reminded by the reviewer, we'd like clarify the reason that why our analytical theory gives imperfect spin values near the boundary in Fig. 3. This is because our analytical theory in the main text is based on the lowest order truncation ($\ell = 0, \pm 1$). If we incorporate more higher orders, namely $\ell = 0, \pm 1, \pm 2, \dots, \pm \ell_{max}$, the pressure field will be generally represented as: $p = \sum_{\ell=-\ell_{max}}^{\ell_{max}} c_{\ell} J_{\ell}(\kappa r) e^{i\ell\varphi} e^{ikz}$. One can then obtain the coefficients c_{ℓ} according to boundary conditions. For the cases considering more higher order ℓ_{max} , we illustrate the theoretical pressure fields $|p|^2$ and spin angular momentum densities s_y in Fig.R1 below. Compared with simulation results, we can see that the analytical theory with the higher order ℓ_{max} will give more accurate results than the lower orders. The SAM values near the boundary will be gradually close to simulation results.

Despite more accurate SAM values for the approximate theory with more higher orders, it will give no more valuable information than the simplest case that taking only $\ell_{max} = 1$. Therefore, in our paper, we took the spirit of Ockham's razor and adopt the simplest approximation with lowest order ($\ell = 0, \pm 1$), which is sufficient to express the SAM values in a neat analytical form, so that clearly reflects our main points: non-zero S_y and tight spin-momentum locking.

Fig. R1 (a) The amplitude of the pressure field $|p|^2$ and spin angular momentum density s_y of the metasurface waveguide eigenmode obtained by numerical simulations. The metasurface boundary is denoted as the yellow color. The $|p|^2$ and s_y obtained theoretically with different ℓ_{max} are shown in (b-e): (b) $\ell_{max} = 2$; (c) $\ell_{max} = 3$; (d) $\ell_{max} = 5$; (e) $\ell_{max} = 7$.

3. In addition to the above comment, Eq. (3) of the analytical theory in the “Method” section is also problematic. The pressure field is decomposed into monopole and dipole modes. However, the wavevectors of the two modes were taken to be equal, while it is known that they have different transverse and longitudinal components of wavevectors in general.

Response:

Thanks for the referee’s concerns about the wave solutions. Please allow us to clarify why there is no problem.

Conventionally, the circular duct waveguide (with the radius R) modes can be decomposed into monopole $J_0(\kappa_0 r)$ and dipole modes $J_{\pm 1}(\kappa_{\pm 1} r)e^{\pm i\ell\varphi}$, corresponding to different eigenmodes with different dispersion relations as shown in Fig. R2(a) (below), $k = \sqrt{\frac{\omega^2}{c^2} - \kappa_\ell^2}$, where the transverse wavevector κ_ℓ will be the solution of $J'_\ell(\kappa_\ell R) = 0$. However, for the meta-surface waveguide with non-symmetric boundary conditions, the eigenmode of our interest will be neither monopole nor dipole modes, but the mixed combinations $\sum_\ell c_\ell J_\ell(\kappa r)e^{i\ell\varphi}$ ($\ell = 0, \pm 1$). And this mode has only a single transverse wavevector κ and a single longitudinal wavevector k with the dispersion as $k = \sqrt{\frac{\omega^2}{c^2} - \kappa^2}$, as shown in Fig. R2(b). The κ will be the first solution of equations: $\sum_\ell c_\ell J'_\ell(\kappa R) = 0$ for hard boundaries and $\sum_\ell c_\ell J_\ell(\kappa R) = 0$ for metasurface (soft) boundaries.

Fig. R2 (a) The dispersions for monopole and dipole modes in the conventional waveguide. The monopole and dipole modes are denoted as the blue and yellow line respectively. For the monopole mode, $\kappa_0 = 0$. (b) The dispersion for a typical mode in the metasurface waveguide. The gray dashed line is the sound cone $k = \frac{\omega}{c}$.

4. As also commented by the other referee, it is important to explain that why the waveguide corners do not induce strong reflections. In the revised manuscript, the authors only explain on Page 4 that this is because the corners “preserves SAM and fail to induce strong backscattering” and “backscattering requires strong scatters that reverse spin”. Why do the “strong” corners preserve spin? The pair of spin states are related by mirror operation and time-reversal operation, i.e., any scatter applies a mirror operation (with mirror plane perpendicular to x or z axis) or time-reversal operation to one spin will transform it into the opposite spin and, therefore, cause reflections. Both the U-shape corner and the T-shape corner apply z-mirror operation on the spin state and, therefore, must induce certain reflection. The reflections in the proposed systems are weak. This might be attributed to the

asymmetric distribution of the waveguide field, i.e., the pressure field mainly localizes on one half of the waveguide cross section without cylinder decorations. The asymmetry of the waveguide field renders it easier to turn left than right in the U-shape case, and easier to couple to the left part than the right part in the T-shape case. If the U-shape waveguide is bended towards the decorated boundary, the reflection should be much stronger. Strong reflection can also happen in the T-shape case if the boundary decoration of the horizontal part faces upward. Therefore, it is fine to state that spin-momentum locking can assist in asymmetric coupling at the corners, but the weak reflection is not due to the near-orthogonality of the two spin states.

Response:

Thanks for the referee's concerns about weak corner-backscatterings. Yes, we agree with the referee. We have modified and rewritten some statements about this part to avoid any confusion.

Actually, there are indeed reflections when the wave passes through the corners. And whether the corners are decorated by the metasurface will induce different reflection behaviors. To clarify these points, we calculate the transmission T of ideal "soft" metasurface waveguides with different bending angles, shown in Fig. R3 (below). The transmissions of bended metasurface waveguides are always better than the conventional circular duct waveguide, especially for $\theta \in [0, \pi/2]$, whose transmission is nearly 100%. This is assisted by the tight acoustic spin-momentum locking of the acoustic spinful mode in the meta-waveguide.

For right-angle bendings $\theta = \pm \frac{\pi}{2}$, the transmissions of the spinful mode in metasurface waveguides will be $\approx 99.6\%$ for $\theta = \frac{\pi}{2}$ and $\approx 74.9\%$ for $\theta = -\frac{\pi}{2}$, both are better than the transmissions of conventional spinless waveguide modes. Indeed, the transmission of metasurface waveguide modes are dependent on the bending angle θ and specially the transmission is not symmetric with respect to θ : $T(\theta) \neq T(-\theta)$. In this point, we agree with the referee that this might be due to the mainly localized field distribution of the eigenmodes on metasurface decorated sides ($|p|^2$ in Fig. 2 and Fig. R1). However, the bending with $\theta < -0.14\pi$ is not accessible for experimental realizations due to the volume interactions of sider bars on metasurfaces shown in Fig.R3(c).

In Fig. R4, for the accessible angles, due to the weak reflections and spin-momentum locking, the forward mode will be held, demonstrating the SAM preservation. The preservation of the mode pattern has the trend to keep the forward propagation so that suppresses the backscattering. For the normal waveguide without metasurface decorations (see Fig. 5a in the main text), the mode pattern will be strongly altered when passing through the right-angle corner, which induces the obvious backscattering.

We thank the referee for his/her deep insight for improving our research work. Although it is impossible for experiments to bend U and T-shape waveguide towards the metasurface boundary without introducing additional defects, we tried to use the simulations with ideal soft surfaces to clarify this point. We have modified and rewritten some statements about this part to avoid any confusion. Hope the referee will be satisfied with our improvements.

Fig. R3 (a) the bended metasurface waveguide with bending angle θ . Two bending cases have been demonstrated for clarifying the bending. The sound hard boundaries and ideal metasurface boundaries with the reflection phase as π are denoted as gray and blue color respectively. The radius is 4 cm; (b) The transmission of the metasurface waveguide (red dotted line) and the conventional waveguide (blue dotted line) as the function of the bending angle θ . The demonstration frequency is $f = 2.9$ kHz. (c) The side bars on metasurface boundaries will have volume overlaps when $\theta < -0.14\pi$, denoted as the gray dashed line in (b).

Fig. R4. (a) The pressure fields p of different metasurface waveguides with bending angles $\theta = \frac{\pi}{2}, \frac{\pi}{3}, \frac{\pi}{6}, 0, -\frac{\pi}{6}, -\frac{\pi}{3}, -\frac{\pi}{2}$. (b) Their corresponding spin angular momentum density s_y .

5. While the spin rotation part is a bit trivial, it can be kept in the main text as a demonstration of spin manipulation phenomenon.

Response:

We are very grateful to the referee for his/her great efforts during this hard time of COVID-19.

REVIEWERS' COMMENTS:

Reviewer #2 (Remarks to the Author):

The authors have well addressed my concerns and comments. I recommend it for publication.